# Perpetration and Victimization in Offline and Cyber Contexts: A Variable- and Person-Oriented Examination of Associations and Differences Regarding Domain-Specific Self-Esteem and School Adjustment

**DOI:** 10.3390/ijerph181910429

**Published:** 2021-10-03

**Authors:** Christoph Burger, Lea Bachmann

**Affiliations:** 1Department of Developmental and Educational Psychology, Faculty of Psychology, University of Vienna, A-1010 Vienna, Austria; lea.bachmann@univie.ac.at; 2Department of Cognition, Emotion, and Methods in Psychology, Faculty of Psychology, University of Vienna, A-1010 Vienna, Austria; 3Department Psychology and Psychodynamics, Karl Landsteiner University for Health Sciences, A-3500 Krems, Austria

**Keywords:** bullying, school violence, cyberbullying, victimization, perpetration, self-esteem, domain-specific self-esteem, psychological adjustment, psychological maladjustment, person-oriented research

## Abstract

Self-esteem has been identified as a predictor of bullying perpetration and victimization, which, in turn, may lead to school adjustment problems. However, findings regarding the direction and strength of these associations have been inconclusive. This study aimed to resolve this by differentiating between offline and cyber contexts and various self-esteem domains. An online sample of 459 adolescents retrospectively completed measures of self-esteem domains and offline/cyber perpetration and victimization, and a subsample of 194 adolescents also completed measures of loneliness and school adjustment. A mediation analysis of bullying-related variables on the effect of self-esteem domains on school adjustment indicated that offline victimization was the only significant mediator. Positive indirect effects were found for social and emotional self-esteem, and negative indirect effects were found for school performance-related self-esteem. Furthermore, person-oriented analyses examined differences in bullying-related roles regarding self-esteem domains, loneliness, and school adjustment. Victim groups showed lower self-esteem in many domains, but cyber victims showed higher body-related self-esteem. Bullies showed lower school performance-related but higher social self-esteem. Both bullies and victims showed lower school adjustment and more loneliness. Implications for theory and practice are discussed, as the findings are relevant for teachers and could be used to develop and deploy more effective anti-bullying programs.

## 1. Introduction

School bullying is a widespread phenomenon that for many students is associated with psychological maladjustment [1]. Studies have shown that self-esteem is an important antecedent of bullying perpetration and victimization [2,3]. However, the cumulative evidence is inconclusive regarding the actual nature, direction, and strength of the underlying associations. One reason for this may be that global self-esteem is too general a construct that may measure differently weighted combinations of various distinct domain-specific facets of self-esteem, depending on the situation and the individual [4,5]. Differentiating between these domain-specific facets of self-esteem may therefore represent one approach to untangle inconsistencies in previous studies. Furthermore, context effects might have influenced previous results, such as whether bullying occurs in offline or cyber contexts [6]. Thus, another starting point would be to measure bullying-related variables separately in offline and cyber contexts. Finally, variable-oriented approaches, which often rely on the assumption of homogeneous samples rather than differentiating between different at-risk groups of students who have adopted bullying-related roles (e.g., bully, victim, bully-victim), could also be at fault [7,8]. This study contributes to the growing literature on self-esteem, bullying, and psychological adjustment by examining the above-mentioned approaches to shed more light on, and at least partially clarify, inconsistencies in the literature.

### 1.1. Defining Bullying Perpetration and Victimization

Bullying among students is a common and persistent problem in schools around the world [1]. It is usually defined as a subcategory of aggressive behavior that is distinct from ordinary, nonrecurring conflicts between students of comparable standing. To be categorized as bullying, first, there must be an imbalance of power between the victim and the bully (either physically, in numbers, or through other means); second, the aggressive behavior must be repeated over an extended period of time; and third, the perpetrator must intend to harm the victim [3,9]. While bullying refers to actively performing acts that meet the above criteria, bullying victimization refers to being the target of such acts. Bullying can take several forms [10], such as physical (punching or hitting someone), verbal (calling someone names or putting them down in front of others), and relational (excluding someone or talking behind their back) bullying.

### 1.2. Differences between Offline and Cyberbullying

The widespread use of social media and the fact that many students can take photos and have access to the Internet at any time with their smartphones have further exacerbated the problem. Bullying that occurs through digital means is referred to as cyberbullying. Most studies have shown that cyberbullying rarely occurs on its own and is often combined with forms of offline bullying [6], suggesting that cyberbullying is just another form of bullying that can be used in existing bullying cases. A recent meta-analysis [11] showed that both cyber and offline perpetration (*r* = 0.39) and cyber and offline victimization (*r* = 0.32) correlated with medium effect sizes. Although cyberbullying is defined by the same criteria involved in offline bullying (power imbalance, repetition, malicious intent), there are some additional factors that might change some of the mechanisms and dynamics in cyberbullying [12].

In contrast to offline bullying, which often occurs at school and during school hours, bullying via digital and mobile technologies can happen everywhere, 24 hours a day, without a break, and without safe places to hide [13]. Digital technologies also tend to lead to a blurring of identities that are usually clearly defined in offline bullying. For example, on the Internet, the bully can act anonymously (e.g., send anonymous messages; call with a suppressed number; post under a pseudonym), assume the victim’s identity (e.g., impersonate the victim by registering accounts with their name, or sending/posting messages in their name), and also disclose sensitive private information about the victim online. Under the guise of anonymity, perpetrators, who usually systematically select lower-status victims in the offline context, also seem to choose youth as victims who are of equivalent or even higher status [14]. Unlike offline bullying, where the perpetrator sees the victim’s reaction and suffering face-to-face, in the online context, they might only see the computer or cell phone screen and it might therefore be more difficult to grasp and empathize with the victim’s concrete situation [15]. On the Internet, posted messages also tend to leave a longer-lasting trace (a spoken sentence is only available to those present at the moment), and often the content remains permanently on the Internet and cannot be deleted by the victim. However, the impact is not only extended regarding exposure time, but also in terms of the number of people who have access to hurtful or embarrassing photos or texts. The audience can quickly expand beyond friends and peers in the classroom, and cyber-bystanders can even cause this material to go viral [16].

All in all, while victimization and perpetration in offline and cyber contexts seem to have many similarities, the context in which bullying occurs may involve different motivations, behavioral dynamics, and effects on students. Therefore, it seems worthwhile to distinguish between these two forms of bullying when assessing differences between bullying-related student groups.

### 1.3. Bullying and Psychological School Adjustment

Both offline and cyberbullying have been shown to have a negative impact on students’ mental health, which also affects their functioning in school [17,18]. These effects are often not limited to adolescence but can extend into adulthood [19]. Bullying is therefore a major threat to public health [20] and to the educational system [21].

There is an ongoing debate as to whether the negative consequences of bullying behavior affect bullies and victims equally. However, empirical studies have shown that many adverse effects on health and school-related outcomes are shared by victims and perpetrators in both offline and cyber contexts. In terms of mental health, both report higher levels of depression [18,22,23,24], anxiety [18,25], and conduct problems [26]. However, perpetrators are more prone to externalizing behavior problems such as aggressive behavior, substance abuse, and delinquency [27,28], while victims are more prone to internalizing problems such as depression, loneliness, and hopelessness [23,26,28,29,30]. Regarding school-related social and educational aspects, both victims and perpetrators reported feeling unsafe at school and experiencing a negative school climate [18,31]. They were also more likely to report higher levels of school absenteeism [18] and showed lower academic achievement [18,31]. 

Students who are both victims and bullies (i.e., bully-victims) are at even higher risk of developing psychological disorders and adjustment problems [32] such as depression and anxiety [33], with students identified as being a combination of offline–online bully-victims having the highest risk factors for depression and somatic symptoms [34].

### 1.4. The Role of Self-Esteem in Bullying and Victimization

Self-esteem refers to the evaluative component of self-knowledge [35]. Because an unfavorable evaluation of the self (i.e., low global self-esteem) may be a direct risk factor for depression [36], it is important to examine potential mediators of this effect. Studies have shown that self-esteem is an important precursor to bullying and victimization [2,3], which in turn may be associated with psychological adjustment problems such as loneliness, anxiety, and depression.

Both offline and cyber victims have been found to display lower levels of self-esteem [22,37,38,39], with particularly low self-esteem among cyber victims [40]. Students with low self-esteem have been described to signal insecurity, which in turn might lead perpetrators to choose them as their victims [2,3]. Victims also seem to be less able to negotiate conflicts that arises in relationships [41] and are more likely to withdraw, rather than defend themselves, making them even easier targets for bullies. When bullied, students with low self-esteem might not be able to take active steps to escape from being victimized [7], potentially leading to increased victimization over time [9]. This systematic victimization might lead to more serious adjustment problems over time [42].

The present study assumes a multiple parallel mediation model [43], in which self-esteem variables are associated with bullying or victimization, which in turn are associated with more severe psychological adjustment problems in school [44]. Although this model is plausible, the underlying mechanisms may be more complex [7], as reciprocal mechanisms such as negative feedback loops may be at play [45]. For example, self-esteem could lead to victimization, which in turn could decrease self-esteem, which again could lead to even more victimization, and so forth. This could eventually lead to even more severe adjustment problems. To date, the complex mechanisms involved are still controversially debated, and more research is needed to shed further light on this subject.

Regarding bullying perpetrators and their relationship with self-esteem, research has also provided conflicting results. While there is empirical evidence that bullies have high self-esteem [38,39,46], there are also studies that support the validity of the classic picture of bullies with low self-esteem [47]. However, it has also been pointed out that these findings may be artifacts, as the differences between bullies and bully-victims were not taken into account in variable-oriented studies [7,8]. A recent meta-analytic review found that self-esteem was, on average, only marginally related to bullying behavior (*r* = 0.07), although associations varied widely across studies (range: −0.51 to 0.33) [48] (p. 191). In summary, the group of students who bully others is quite diverse, but most research has not taken this heterogeneity into account [8]. At the very least, it seems important to distinguish between pure bullies and bully-victims.

### 1.5. Distinguishing between Different Functional Domains of Self-Esteem

While global self-esteem refers to how individuals evaluate themselves as a whole, domain-specific facets of self-esteem refer to how they evaluate themselves in various relevant life domains. Such functionally distinct facets of self-esteem might include self-perceptions in social, school performance-related, body-related, creative-artistic, and emotional domains [4], and have been shown to predict outcomes related to psychological adjustment, health, education, and work [49,50,51]. Self-esteem domains have been shown to be as comparably stable as global self-esteem [52].

Global self-esteem can be understood as an index of various facets of more specific self-esteem domains. The degree to which specific self-esteem facets determine global self-esteem was found to be a function of how highly the specific facet is personally valued [53], but more research is needed to better understand these complex interrelations [54]. In other words, the weighting of different domain-specific facets in this index might differ substantially across individuals, and thus global self-esteem might measure different constructs in different individuals.

### 1.6. The Role of Domain-Specific Self-Esteem in Bullying and Victimization

Using measures of domain-specific self-esteem facets instead of global self-esteem might lead to a more comprehensive understanding of the associations with bullying perpetration and victimization, as different self-esteem facets might be differently associated with particular bullying-related behaviors or roles [55]. Thus, inconsistent findings regarding the association between self-esteem and bullying behavior might be, at least in part, explained by distinguishing between different facets of self-esteem [5,7].

Low levels of self-esteem in the social domain (i.e., self-perceived peer social competence) have been repeatedly identified as a risk factor for victimization [2,56,57,58,59,60]. Some studies also found this association with perpetration [7,56,57,61,62,63], but other studies did not [5,64,65]. In some studies, high levels of social self-esteem were associated with (verbal) perpetration [66]. Some perpetrators have been reported to have higher social self-esteem than their peers, suggesting that they overestimate their social connectedness [67] and that their bullying behavior may be a response to their positive social self-esteem being threatened by peers who reject them [68].

Most previous studies also found a negative relationship between school performance-related self-esteem and victimization [56,57,58,59,60,65], though positive relationships were also found [69]. Low school performance-related self-esteem was also found to be associated with perpetration [5,56,57,63,69].

Furthermore, low scores on body- or appearance-related self-esteem domains might also be related to victimization [56,58,59,60]. In obese students, this association might be particularly high [70,71]. Perpetration, however, was not found to be associated with lower body-related self-esteem [5,56,58,65].

High levels of emotional self-esteem were also associated with perpetration [65,69]. There were no studies examining differences in creative-artistic self-esteem; however, in one study, bully-victims were shown to exhibit higher levels of creativity than victims and non-involved students [72].

Many of the above studies did not take into account the fact that perpetration and victimization may co-occur in some students (bully-victims). Since it has been suggested that bullies might have higher levels and bully-victims have lower levels of self-esteem, a failure to distinguish between pure perpetration, pure victimization, and their co-occurrence would confound results. Some studies which included bully-victims [57,65,73] found that they had lower levels of social self-esteem than bullies, victims, and non-involved students, while others found that their ratings were more similar to those of victims than bullies [56,58]. Regarding school performance-related self-esteem, bully-victims have been found to have lower scores compared to both victims and bullies in some studies [56,65], while in other studies their scores were more similar to those of bullies than to those of victims [58]. In comparison to bullies, they have also been shown to have lower levels of perceived body-related self-esteem (physical attractiveness) [65]. Regarding emotional self-esteem, bully-victims have been found to score lower than both victims and bullies [65].

In sum, it seems to be useful to distinguish between different facets of self-esteem when examining relationships with bullying-related behavior. It also seems advisable to combine variable-oriented approaches that examine associations between bullying-related variables and other constructs of interest with person-oriented approaches [74] that also explore differences between different bullying-related role groups, including bully-victims.

### 1.7. Short-Term Retrospective Measurement of Bullying-Related Behavior

Having students self-report their current bullying-related behaviors could present several challenges. First, students who are currently actively harassing other students or being harassed by other students may be reluctant to admit it [75], perhaps because they are ashamed of their predicament or fearful of punishment. Being asked about ongoing bullying incidents may also trigger uncomfortable thoughts or feelings (cognitive dissonance, emotional distress [76]) that could have serious ethical implications [77] and might not be conducive to data validity. In addition, asking younger students about bullying behaviors may be complicated by their limited cognitive development (especially when answering longer questionnaires with more complex items [78]) and limited moral cognitive development [79].

We suggest that a useful alternative might be to have adolescents who have recently completed their school years (ages 18 to 25) report retrospectively on bullying-related behaviors during their school years. Information about one’s undesirable behavior or experiences in the past might be perceived as less self-threatening because of the time gap and the fact that it is not directly related to the current self. Participants might still have rich memories of their school years due to the close temporal distance, avoiding more severe memory biases that might be associated with long-term reports [80], although retrospective reports of adverse childhood experiences have been found relatively valid even after 40 years, such as [81]. At the end of their school careers, students may have developed more consistent memories than they might have had during their potentially turbulent school years [82]. Retrospective, short-term surveys could also allow for summative assessments that span over longer periods of time, build on a broader perspective, and benefit from a deeper and more mature understanding of social behavior that participants have developed over time [82] (compared to snapshot estimates that might be more affected by noise such as daily fluctuations). Because of their advanced cognitive abilities, such as perspective taking and moral reasoning [79], school graduates also have a deeper understanding of survey items than their younger peers [78]. From a practical perspective, another advantage is that data collection with adolescents of legal age can be easily conducted online, as parental consent does not need to be obtained. All in all, we suggest that, just as self-reports and peer-reports of bullying behavior have been found to complement each other [83,84,85], retrospective self-reports may also allow researchers to measure facets that may not be gauged in concurrent self-reports (and vice versa).

### 1.8. Research Questions of the Present Study

So far, no studies have examined whether bullying victimization and perpetration in offline and cyber contexts jointly mediate the relationship between different facets of self-esteem and psychological adjustment in school. Therefore, the purpose of the present study aimed to fill this research gap by computing a multiple parallel mediation model [43]. Based on previous research, we expected self-esteem domains to be positively related to school adjustment. Furthermore, we expected indirect effects via bullying-related behavior variables. Although the topic is still controversial, many studies have found negative effects of both victimization and perpetration on psychological adjustment. We thus anticipated negative effects for both victimization and perpetration on school adjustment. Following the results of studies suggesting that cyberbullying may, in most cases, be an extension of already ongoing bullying behavior [6], we anticipated that only offline bullying-related behavior would be negatively associated with psychological school adjustment. Regarding differential associations of self-esteem domains on bullying-related behavior, following the literature reviewed above, we expected a negative association of social and school performance-related self-esteem with both perpetration and victimization. Furthermore, we anticipated body-related self-esteem to be positively associated with perpetration and negatively associated with offline (but not online) victimization. Finally, we hypothesized that emotional self-esteem would be positively associated with perpetration. We had no explicit hypothesis for creative-artistic self-esteem.

Secondly, previous research has pointed to the importance of taking a person-oriented approach that examines differences between groups of students who have adopted different bullying-related roles. One advantage of this approach is that it can easily differentiate between bullies, victims, and bully-victims. The present study aimed to extend this technique by also including the mode of perpetration or victimization, that is, offline or cyber, in the classification of bullying-related groups. After determining which groups can be found empirically in the present sample, our study sought to extend the current literature by exploring differences between these bullying-related roles in various measures of self-esteem, psychological school adjustment, and loneliness.

## 2. Materials and Methods

### 2.1. Participants

A total of 504 German-speaking young people completed an online questionnaire from December 2020 to May 2021. A minimum age of 18 years and a maximum age of 25 years were set as inclusion criteria because the participants should have completed their school years in order to be able to assess them retrospectively in their entirety, but the school years should also not be too far back in the past in order to still ensure reliable recall. A total of 19 people did not meet these criteria. Another 26 persons only filled out the first page of the questionnaire. Individuals who left parts of the questionnaire unanswered were left in the data set because full information maximum likelihood estimators were used to perform the multiple parallel mediation analyses.

Thus, the final sample included 459 individuals (79.1% female) between the ages of 18 and 25 (*M* = 22.98, *SD* = 1.86). The average duration of questionnaire completion was 9.08 min (*SD* = 2.87). Regarding the highest level of education completed, 49.0% of participants reported having obtained a bachelor’s degree, 45.1% reported having obtained a secondary school diploma (equivalent to general university entrance qualification), 2.2% reported having obtained a master’s degree, 1.7% reported having completed vocational training (including apprenticeship or vocational school), 1.7% reported having completed compulsory elementary education, and the remaining 0.2% reported having obtained a doctoral degree.

Two constructs, psychological school adjustment and loneliness, were added to the online questionnaire at a later time point, thus data on these constructs were collected with a smaller portion of the sample, consisting of *n* = 194 participants (77.3% female) aged 18 to 25 years (*M* = 22.38, *SD* = 2.15).

### 2.2. Procedure

Data collection took place cross-sectionally at one point in time by means of an online questionnaire, which was advertised primarily via online social networks. Participants were informed about the key facts of the survey on the first page (e.g., inclusion criteria, topic areas queried, expected duration, voluntariness of participation, data protection, expected benefits and risks) and had to give informed consent to complete the survey. Participation was not compensated. In addition to changes in item wording, each page of the online questionnaire clearly stated that the questions related to school years and should be completed retrospectively to the best of the respondent’s knowledge.

### 2.3. Measures

After giving informed consent, participants were asked to provide their gender (0 = *female*, 1 = *male*), age, and previous education. Furthermore, the following six constructs were measured. Reliabilities, means, standard deviations, and bivariate correlations are presented in Table 1.

#### 2.3.1. Cyber Victimization and Perpetration

The cyberbullying test [86] was translated into German and used to assess cyber victimization and cyber perpetration. Participants retrospectively reported the average frequency with which they engaged in and were targeted by 15 different behaviors during their school years (answer options: 0 = *never*, 1 = *sometimes*, 2 = *quite a few times*, 3 = *always*). Before answering the items, participants read a brief instruction explaining the retrospective measure (e.g., “they” can refer to single or multiple persons) and that it relates to their school years. Sample items for cyber victimization are: “Have they ever sent you offensive and insulting messages by cellphone or Internet?” and “Have they ever taken pictures of you without your permission in places such as locker rooms, beaches, or toilets and put them on the Internet or diffused them by cellphone?”. Sample items for cyber perpetration are: “Have you ever sent offensive and insulting messages to someone by cellphone or Internet?” and “Have you ever taken pictures of someone without their permission in places such as locker rooms, beaches, or toilets and put them on the Internet or diffused them by cellphone?”

#### 2.3.2. Offline Victimization and Perpetration

A self-constructed measure similar to the cyberbullying test [86] was used to measure offline victimization and offline perpetration. Participants retrospectively reported the average frequency with which they engaged in and were targeted by four different behaviors during their school years (answer options: 0 = *never*, 1 = *sometimes*, 2 = *quite a few times*, 3 = *always*), including physical, verbal, relational, and property-related forms of bullying, for example see [10]. Before answering the items, participants read a brief instruction, explaining the retrospective measure and that it refers to their school years. Sample items for offline victimization were: “Classmates have harassed me verbally in class (called me names, yelled at me, laughed at me, etc.).”, and “Classmates have harassed me physically in the classroom (pushed, pushed, hit me, etc.).”. Sample items for offline perpetration were: “I have excluded other classmates from activities in the classroom (not let them play, ignored them, etc.).”, and “I have mistreated other classmates’ property (taken away personal items and damaged them, etc.).”

#### 2.3.3. Global Self-Esteem

An adapted version of the German version of the Rosenberg self-esteem scale [87] was used to measure global self-esteem. It is a 10-item 4-point Likert scale with answer options ranging from *strongly disagree* (*1*) to *strongly agree* (4). Participants read a brief introduction indicating that the items related to their school years. Sample items are “On the whole, I was satisfied with myself.” and “I felt that I had a number of good qualities.”

#### 2.3.4. Domain-Specific Self-Esteem

The EMES-16 [4] was translated into German and adapted to retrospectively measure five domain-specific facets of self-esteem. It is a 16-item 5-point Likert scale with answer options ranging from *strongly disagree* (1) to *strongly agree* (5). Subscales measure emotional self-esteem (3 items), social self-esteem (3 items), body-related self-esteem (3 items), school performance-based self-esteem (4 items), and creative-artistic self-esteem (3 items).

#### 2.3.5. Psychological School Adjustment

Three 7-point Likert Items reported by Kochenderfer-Ladd [88] were translated into German and extended to 9 items to capture psychological school adjustment. The items were adapted to refer retrospectively to the school years. Answer options ranged from *completely disagree* (1) to *completely agree* (7). Sample items are “When I was in school, I was happy.”, and “When I was in school, I worried.” After scoring the items, higher values represent higher levels of adjustment.

#### 2.3.6. Loneliness

The 6-item De Jong Gierveld loneliness scale [89] was used in order to measure both social (3 items) and emotional (3 items) loneliness. Items were 5-point Likert scales (answer options: 1 = *yes!*, 2 = *yes*, 3 = *more or less*, 4 = *no*, 5 = *no!*). Items were translated into German and adapted in order to retrospectively refer to the school years. A sample item for social loneliness was “There were plenty of people I could rely on when I had problems.” and a sample item for emotional loneliness was “I experienced a general sense of emptiness.”

### 2.4. Data Analytical Strategy

#### 2.4.1. Factor Structure of Domain-Specific Self-Esteem Items

Because the items have been translated into German and were substantially adapted to retrospectively measure self-esteem in a school setting, we used exploratory factor analysis (EFA) with an oblique rotation (promax [90]) to establish factor structure. Bartlett’s test of sphericity and the Kaiser–Meyer–Olkin measure were used to establish the suitability of the data for factor analysis. Empirical EFA eigenvalues were compared to the 95th percentile eigenvalues (obtained in a parallel analysis with Monte Carlo simulated data) as a cut-off criterion for determining the number of eigenvalues that are significantly different from those expected from random data [43].

#### 2.4.2. Mediation Model

Bivariate associations between study variables were calculated before including them in the more complex model. A parallel multiple mediation model was then calculated using the statistics program JASP version 0.14.1.0 [43]. Self-esteem domains were included as predictors, offline and cyber victimization, as well as offline and cyber bullying as mediators, and psychological school adjustment as outcome, while controlling for gender and age. A full information maximum likelihood estimator was used with robust standard errors that could also handle missing values. Direct, indirect, total indirect, and total effects were calculated.

#### 2.4.3. Construction of Bullying-Related Role Groups

Using a person-oriented approach, bully-related groups were formed, including all combinations of both offline and cyber victimization as well as offline and cyber bullying. With respect to offline perpetration, cyber perpetration, offline victimization, and cyber victimization scales, the students who selected the response option *often* or *always* (equivalent to a response score > 1) for at least one of the items were identified [86]. These students were labeled offline perpetrators, cyber perpetrators, offline victims, or cyber victims. Next, it was observed which groups were empirically present and, based on this information, new groups were formed that were mutually exclusive (i.e., a student could only be a member of one group). Finally, the prevalence of these groups was calculated.

#### 2.4.4. Differences between Bullying-Related Roles

A series of ANCOVAs was conducted to test for differences between different bullying-related behavior roles (non-involved students, pure bullies, pure offline victims, pure cyber victims, pure offline-cyber victims, bully-victims) on different self-esteem facets, psychological school adjustment, and loneliness, controlling for gender and age. With the variables psychological school adjustment and loneliness, a smaller subsample (*n* = 194) was used. Because all bully-victims in this subsample were male, analyses in this subsample could only control for age but not gender.

## 3. Results

### 3.1. Domain-Specific Facets of Self-Esteem

Bartlett’s test of sphericity was significant, χ^2^ (120) = 2793.14, *p* ≤ 0.001, and Kaiser–Meyer–Olkin measure of sampling adequacy was 0.82, indicating that the present data were suitable for factor analysis. Parallel analysis suggested a five-factor solution (see Appendix A): (1) emotional self-esteem, consisting of three items, (2) school performance-related self-esteem, consisting of four items, (3) creative-artistic self-esteem, consisting of three items, (4) social self-esteem, consisting of three items, (5) body-related self-esteem, consisting of three items. Detailed information on the scales and factor loadings can be found in Table 2.

### 3.2. Bivariate Associations

Bivariate relations among the main study variables are shown in Table 1. Almost all self-esteem measures intercorrelated positively with small to large effect sizes. Emotional and social loneliness intercorrelated positively with a large effect size, and the same was true for offline and cyber perpetration and for offline and cyber victimization. Interestingly, cyber victimization and cyber perpetration were positively correlated.

Regarding correlations across constructs, psychological school adjustment correlated positively with all self-esteem measures except creative-artistic self-esteem, and negatively with emotional and social loneliness, offline and cyber victimization, and cyber (but not offline) perpetration. Many self-esteem measures were negatively correlated with emotional and social loneliness, and with measures of offline and cyber perpetration and victimization. Finally, emotional, and social loneliness were also correlated positively with some of the measures of offline and cyber perpetration and victimization.

### 3.3. Mediational Role of Offline and Cyber Bullying and Victimization

A multiple parallel mediation model (see Figure 1) was used to determine path coefficients (Appendix A) and total, direct, indirect, and total indirect unstandardized and standardized effects (see Appendix A). The model revealed significant positive total effects of school performance-related, social, and emotional self-esteem on psychological school adjustment, which were all partially mediated. Body-related and creative-artistic self-esteem had neither significant direct nor indirect effects on school adjustment.

Social self-esteem had a positive total indirect effect (γ = 0.131, *p* < 0.01) and a positive indirect effect (γ = 0.153, *p* < 0.01) via offline victimization on school adjustment. Emotional self-esteem had a positive total indirect effect (γ = 0.050, *p* = 0.05) but no significant single indirect effects on school adjustment. School performance-related self-esteem had a negative indirect effect (γ = −0.082, *p* = 0.01) on school adjustment via offline victimization but had not significant total indirect effect.

### 3.4. Prevalence of Bullying-Related Roles

All individuals who had selected *often* on at least one item of the respective scale (offline perpetration, offline victimization, cyber perpetration, cyber victimization) were assigned to the mutually non-exclusive preliminary bullying-related roles [86]. For the total sample, 4.8% were offline perpetrators, 17.9% were offline victims, 2.0% were cyber perpetrators, and 15.0% were cyber victims. All individuals who were not assigned to any of these bullying-related groups were classified as “noninvolved students” (74.7%).

Preliminary bullying-related roles were crossed to yield all logically possible mutually exclusive role combinations. In the total sample, the following mutually exclusive preliminary groups were found: (a) 328 non-involved, (b) 10 pure offline perpetrators, (c) 1 pure cyber perpetrator, (d) 2 pure offline-cyber perpetrators, (e) 29 pure offline victims, (f) 16 pure cyber victims, (g) 43 pure offline-cyber victims, (h) 1 offline perpetrator offline victim, (i) 1 cyber perpetrator cyber victim, (j) 1 offline-cyber bully cyber victim, (k) 4 offline perpetrators offline-cyber victims, and (l) 3 offline-cyber bully offline-cyber victims.

Because some group combinations were very small, they were consolidated, resulting in the following six groups: (a) noninvolved students, (b) pure perpetrators (offline and/or cyber perpetrators), (c) pure offline victims, (d) pure cyber victims, (e) pure offline-cyber victims, and (f) bully-victims (offline and/or cyber perpetrators and offline and/or cyber victims). At least two-thirds (66.6–74.7%) of participants were non-involved. Victims were the next largest group, with approximately 20.0–29.1% (including 28.6–32.9% offline only, 16.1–18.2% cyber only, and 48.9–55.3% offline-cyber). The two smallest groups were bullies with 2.6–3.0% and bully-victims with 1.6–2.3%. More detailed information on the prevalence and demographics of the groups can be found in Table 3.

### 3.5. Bulling-Related Group Differences in Self-Esteem, Loneliness, and School Adjustment

**Global Self-Esteem.** ANCOVA revealed a significant effect of bullying-related behavioral roles on global self-esteem after controlling for gender and age (see Table 4). Tukey post hoc tests revealed significant mean differences between noninvolved students (*M* = 3.09, *SD* = 0.55) and the following two groups: pure offline victims (*M* = 2.58, *SD* = 0.60, *p* ≤ 0.001, *d* = 0.86), and pure offline-cyber victims (*M* = 2.57, *SD* = 0.60, *p* ≤ 0.001, *d* = 0.91).

**Social Self-Esteem.** ANCOVA revealed a significant effect of bullying-related behavioral roles on social self-esteem after controlling for gender and age (see Table 4). Tukey post hoc tests revealed significant mean differences between noninvolved students (*M* = 3.85, *SD* = 0.80) and the following two groups: pure offline victims (*M* = 2.89, *SD* = 0.90, *p* ≤ 0.001, *d* = 1.15) and pure offline-cyber victims (*M* = 2.72, *SD* = 0.97, *p* ≤ 0.001, *d* = 1.32). Further significant differences were found between pure bullies (*M* = 4.08, *SD* = 1.06) and the same two victim groups: pure offline victims (*M* = 2.89, *SD* = 0.90, *p* ≤ 0.01, *d* = 1.17) and pure offline-cyber victims (*M* = 2.72, *SD* = 0.97, *p* ≤ 0.001, *d* = 1.27). All the other groups did not differ significantly.

**School Performance-Related Self-Esteem.** ANCOVA revealed a significant effect of bullying-related behavioral roles on school performance-related self-esteem after controlling for gender and age (see Table 4). Tukey post hoc tests revealed a significant mean difference between pure bullies (*M* = 2.79, *SD* = 1.36) and the following two groups: noninvolved students (*M* = 3.81, *SD* = 0.85, *p* ≤ 0.01, *d* = 1.05) and pure offline-cyber victims (*M* = 3.92, *SD* = 0.72, *p* ≤ 0.01, *d* = −1.12). All the other groups did not differ significantly.

**Body-Related Self-Esteem.** ANCOVA revealed a significant effect of bullying-related behavioral roles on body-related self-esteem after controlling for gender and age (see Table 4). Tukey post hoc tests revealed significant mean differences between noninvolved students (*M* = 3.00, *SD* = 0.97) and the following two victim groups: pure offline victims (*M* = 2.33, *SD* = 0.92, *p* = 0.02, *d* = 0.66) and pure offline-cyber victims (*M* = 2.24, *SD* = 0.92, *p* ≤ 0.001, *d* = 0.73). Further significant differences were found between pure cyber victims (*M* = 3.72, *SD* = 1.11) and the following three groups: pure offline victims (*M* = 2.33, *SD* = 0.92, *p* ≤ 0.001, *d* = 1.41), pure offline-cyber victims (*M* = 2.24, *SD* = 0.92, *p* ≤ 0.001, *d* = 1.43), and bully-victims (*M* = 2.22, *SD* = 1.50, *p* ≤ 0.01, *d* = 1.18). All the other groups did not differ significantly.

**Emotional Self-Esteem.** ANCOVA revealed a significant effect of bullying-related behavioral roles on emotional self-esteem after controlling for gender and age (see Table 4). Tukey post hoc tests revealed significant mean differences between noninvolved students (*M* = 3.49, *SD* = 0.91) and the following two groups: pure offline victims (*M* = 2.61, *SD* = 1.03, *p* ≤ 0.001, *d* = 0.89) and pure offline-cyber victims (*M* = 2.70, *SD* = 0.93, *p* ≤ 0.001, *d* = 0.84). All the other groups did not differ significantly.

**Creative-Artistic Self-Esteem.** ANCOVA revealed no significant effects of bullying-related behavioral roles on creative-artistic self-esteem after controlling for gender and age (see Table 4).

**Psychological Adjustment in School.** ANCOVA revealed a significant effect of bullying-related behavioral roles on psychological school adjustment after controlling for age (see Table 4). Tukey post hoc tests revealed significant mean differences between noninvolved students (*M* = 5.11, *SD* = 1.20) and the following groups: pure bullies (*M* = 3.18, *SD* = 1.85, *p* ≤ 0.008, *d* = 1.59), pure offline victims (*M* = 3.22, *SD* = 1.25, *p* ≤ 0.001, *d* = 1.57), and pure offline-cyber victims (*M* = 3.18, *SD* = 1.05, *p* ≤ 0.001, *d* = 1.66). All the other groups did not differ significantly.

**Emotional Loneliness.** ANCOVA revealed a significant effect of bullying-related behavioral roles on emotional loneliness after controlling for age (see Table 4). Tukey post hoc tests revealed significant mean differences between noninvolved students (*M* = 2.43, *SD* = 0.90) and the following groups: pure offline victims (*M* = 3.67, *SD* = 1.10, *p* ≤ 0.001, *d* = 1.34) and pure offline-cyber victims (*M* = 3.56, *SD* = 0.90, *p* ≤ 0.001, *d* = 1.24). All the other groups did not differ significantly.

**Social Loneliness.** ANCOVA revealed a significant effect of bullying-related behavioral roles on social loneliness after controlling for age (see Table 4). Tukey post hoc tests revealed significant mean differences between noninvolved students (*M* = 2.05, *SD* = 0.94) and the following groups: pure bullies (*M* = 3.67, *SD* = 1.45, *p* ≤ 0.006, *d* = −1.67), pure offline victims (*M* = 3.58, *SD* = 1.20, *p* ≤ 0.001, *d* = −1.57), and pure offline-cyber victims (*M* = 3.48, *SD* = 1.07, *p* ≤ 0.001, *d* = −1.47). All the other groups did not differ significantly.

## 4. Discussion

This study is the first to retrospectively examine the mediation effects of offline and cyber victimization, as well as offline and cyber perpetration on the association between five domain-specific self-esteem facets and psychological school adjustment, while controlling for gender and age. This approach allowed us to identify the distinct effects of self-esteem domains, but also to distinguish between offline and cyber behaviors. Furthermore, this study employed a person-oriented approach, examining differences in global and various domain-specific facets of self-esteem between six different bullying-related groups, including non-involved students, offline victims, cyber victims, and offline-cyber victims, bullies, and bully-victims.

### 4.1. Bivariate Associations between Victimization and Perpetration in Offline and Cyber Contexts

As expected, offline victimization correlated highly with cyber victimization (*r* = 0.69, *p* < 0.001), and offline perpetration correlated highly with cyber perpetration (*r* = 0.65, *p* < 0.001). These findings are in line with studies suggesting a large overlap between these two modalities [6], but are higher than those reported in a recent meta-analysis [11]. Interestingly, offline perpetration was not correlated with either form of victimization, and offline victimization was not correlated with either form of perpetration. Cyber perpetration and cyber victimization were, however, correlated (*r* = 0.18, *p =* 0.01), possibly confirming previous results that suggest that role changes (e.g., cyber perpetration as a reaction of previous cyber victimization [92]) might be more prevalent in cyber contexts.

### 4.2. Multivariate Associations of Self-Esteem Domains and Bullying-Related Behavior

The first interesting result pertains to the association of self-esteem domains with bullying-related behavior. While all self-esteem domains except creative-artistic self-esteem showed some bivariate correlations with bullying-related behavior, only social and school performance-related self-esteem emerged as significant predictors in the multivariate analysis. This was unexpected because previous studies have suggested that victimization was negatively associated with low body-related self-esteem [56,58], and that bullies tend to exhibit higher levels of emotional self-esteem [56,69].

Secondly, only victimization, but not perpetration (irrespective of offline or cyber), was predicted by self-esteem domains. Thus, with respect to the debate about whether bullying perpetration is related to self-esteem [8], the results of the present study seem to negate this. However, it must be noted that the present analysis did not address any interaction effects (e.g., victimization–perpetration co-occurrence).

### 4.3. Multivariate Associations of Bullying-Related Behavior with Psychological School Adjustment

Although bivariate correlations of cyber (but not offline) perpetration, as well as offline and cyber victimization, revealed negative associations with psychological school adjustment, in multivariate analysis, only offline victimization predicted psychological adjustment. This can be interpreted as confirming the view that only victimization but not perpetration is uniquely contributive to school adjustment problems when analyzed jointly, and might be taken as support for the notion that cyber bullying is often not a unique form of bullying, but rather an extension of already existing bullying episodes [6].

### 4.4. Total and Indirect Effects of Self-Esteem Domains on School Adjustment

Social, school performance-related, and emotional self-esteem had positive total effects of school adjustment, thus they can be regarded as protective factors. Body-related and creative self-esteem were uncorrelated.

This study extended the existing literature by revealing that the association of self-esteem domains on school adjustment was partially mediated by bullying-related behavior. Social self-esteem had a positive indirect effect on school adjustment via offline victimization and a positive total indirect effect via all of the bullying-related behaviors. Emotional self-esteem also showed a positive total indirect effect via all of the bullying-related behaviors. School performance-related self-esteem, however, had a negative indirect effect on school adjustment via offline victimization. These results might be of practical relevance for developers of anti-bullying programs or for teachers [93], as students with high levels of school performance-related self-esteem and low levels of social self-esteem might be at risk for offline victimization, and students with low levels of social, school performance-related, an emotional self-esteem might be at risk for adjustment problems.

### 4.5. Differences in Self-Esteem Scores

A person-oriented approach allowed the investigation of differences in global self-esteem and various self-esteem domains. Interestingly, creative-artistic self-esteem was the only self-esteem scale that did not differ among any of the bully-related groups.

Offline victims and offline-cyber victims had very similar scores and both differed from non-involved students in all scales except school performance-related and creative-artistic self-esteem. They had lower global, social, body-related, and emotional self-esteem. Cyber victims had higher body-related self-esteem scores than all the other three victimized student groups (offline victims, offline-cyber victims, and bully-victims). Regarding all the other self-esteem scales they did not differ significantly from non-involved students, nor from the other victims and bullies.

Bullies did not differ from non-involved students in most self-esteem scales with the only exception of having lower school performance-related self-esteem. Bullies also had lower school performance-related self-esteem than offline-cyber victims. They showed, however, higher social self-esteem than both offline victims and offline-cyber victims. Bully-victims also did not differ from non-involved students in any self-esteem scales. They only differed from cyber victims in having lower body-related self-esteem (descriptively, they had the lowest score in body-related self-esteem of all groups).

In summary, participants with offline victimization (offline victims, offline-cyber victims) had lower scores than non-involved students in many self-esteem domains—which is in line with several other studies that identified low levels of social and body-related self-esteem as a risk factor for victimization [56,57,58,59,60]. One exception was school performance-related self-esteem, where they had similar scores as non-involved students but higher scores than bullies. This finding suggests that it is the perpetrators, rather than the victims, that do not consider themselves to be successful in school. Descriptively, the offline-cyber victims had the highest scores in school performance-related self-esteem, suggesting that being particularly successful in school could even be one of the reasons for being selected as a victim (“nerd”, “teacher’s pet”) by less successful students. This result is in contrast to previous studies indicating that victimized students tend to perform worse in school [31] and have low levels of school performance-related self-esteem [56,57,58,59,60]. This finding may be related to the fact that the current study involved highly educated individuals who retrospectively reported on their school years. In other words, individuals who had experienced problems in school due to victimization and dropped out of the educational system as a result are potentially underrepresented in the present sample.

A further interesting finding was that students who were only victimized in the cyber context did not differ in any self-esteem domains from non-involved students, and were especially characterized by having a higher body-related self-esteem compared to offline victims, offline-cyber victims, and even bully-victims. This could be taken as an indication that students who evaluate themselves as physically stronger tend to be bullied online. Cyberbullying could therefore be seen as a kind of alternative option when offline bullying is not possible or too risky for the perpetrator due to physical inferiority [14,94].

### 4.6. Differences in School Adjustment and Emotional and Social Loneliness

Offline victims and offline-cyber victims had more unfavorable scores in all three scales measuring school adjustment and loneliness compared to non-involved students. Bullies showed lower school adjustment and lower social (but not emotional) loneliness than non-involved students. Cyber victims and bully-victims did not differ from non-involved students, other victims, or bullies in any of the three scales.

Summing up, these results again suggest that victims who are victimized offline (offline victims, offline-cyber victims) suffer the most emotionally from the consequences of bullying. One could interpret the results as implying that offline (rather than cyber) victimization is a general indicator that more serious bullying victimization is occurring. Cyberbullying, on the other hand, is more likely an add-on to existing victimization [6], most probably because the online world has become an integral part of students’ daily lives and interactions over the past decades. Therefore, it is not surprising that offline bullying also leads to cyberbullying over time [18]. Previous studies have shown that students who are bullied both in offline and cyber contexts (i.e., poly-victims) might suffer more than those who are only bullied offline [95]; however, the present findings do not confirm this, as these groups show relatively similar outcomes. Furthermore, similar to self-esteem problems, pure cyber victims do not seem to be as affected as other victims regarding adjustment problems and loneliness.

However, in terms of school adjustment and social loneliness, bullies were equally affected as offline or offline-cyber victims, confirming that it is not only victims that show signs of maladjustment [27,28]. Interestingly, in contrast to previous findings in which bully-victims are often portrayed as the most affected students [32], bully-victims in our sample did not have the highest scores from a descriptive standpoint, nor did they differ significantly from non-involved students—although it should be noted here that the number of individuals in this group was small.

Our findings also show that it makes sense to evaluate existing data with a person-oriented approach that identify highly affected groups of students (rather than focus on associations in supposedly homogeneous samples), as these evaluations provide additional information and insights as compared to variable-oriented analyses. These results confirm the view that variable- and person-oriented approaches complement each other [74].

### 4.7. Limitations and Future Directions

The present study is not without limitations. One constraint of the study was its cross-sectional nature. It is clear that the mediational model is a simplified depiction of reality because bi-directional reciprocal effects might be at play [45], and self-esteem as well as school adjustment might be both predictors and consequences of bullying-related behaviors. Furthermore, although the overall sample was large, some variables were only collected in a smaller subsample. Although we expected that participants would be more willing to admit past than currently ongoing perpetration and victimization behaviors, this might have only been the case for victimization but not perpetration, as the numbers of self-reported bullies and bully-victims were low (about 2–3%). This led to relatively small cell sizes and must be considered when interpreting the results. Finally, the sample consisted of German-speaking adolescents (aged 18 to 25 years) with a high level of education (about 50% had academic degrees) and, thus, results might not be generalizable to other populations. Although we did not measure socioeconomic status directly, given the high educational attainment of our participants, we cannot rule out the possibility that individuals with lower socioeconomic status, which has been shown to be a risk factor for both victimization [96] and lower self-esteem [97], are underrepresented in the current sample. Although social media penetration among Austrian youth is nearly 100% [98], the fact that the data collection took place online and was advertised on online social networks may have biased the results, as those who do not use social networks had a lower chance of being included.

Future studies should replicate the current findings in larger longitudinal studies with individuals from different age groups and with different socioeconomic, cultural, and educational backgrounds. Exploring gender differences in more detail might also be of interest. Furthermore, it would be of interest to explore the effect of social desirability in retrospective reporting of undesirable behaviors (such as being a bully, bully-victim, or victim) in more detail. Complementing the measurement instruments with peer- and teacher-reports, and observation-based measures regarding bullying-related behaviors and with implicit measures of self-esteem domains that are harder to fake (e.g., initial preference task or implicit association task [99,100,101]) could also help mitigate such biases.

## 5. Conclusions

The results of the present study suggest that the association between domain-specific self-esteem and school adjustment is partially mediated by bullying-related behavior. Social self-esteem had a positive indirect effect on school adjustment via offline victimization. Both social and emotional self-esteem had a positive total indirect effect via the combination of all bullying-related behaviors. In contrast, school performance-related self-esteem had a negative indirect effect on school adjustment via offline victimization. Victim groups tended to exhibit lower self-esteem in many domains, but cyber victims showed higher body-related self-esteem. Bullies showed lower school performance-related self-esteem but higher social self-esteem. Both bullies and victims showed lower levels of school adjustment and higher levels of loneliness. Pure cyber victims did not seem to be as affected as other victims. In a similar vein, and contrary to previous findings, bully-victims did not differ significantly from non-involved students. Taken together, these findings have practical implications for the improvement of anti-bullying interventions as well as for teachers.

## Figures and Tables

**Figure 1 ijerph-18-10429-f001:**
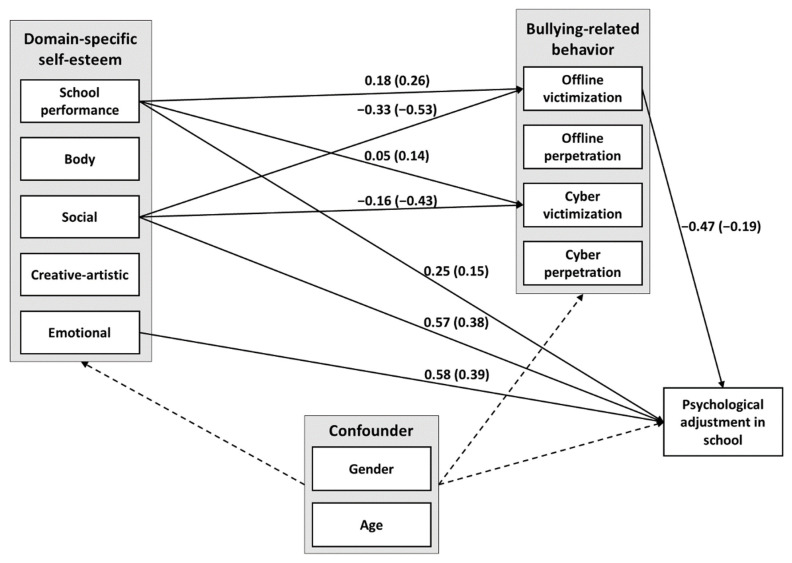
Parallel Multiple Mediation Model: The Effect of Domain-Specific Self-Esteem Facets on Psychological School Adjustment with Offline and Online Perpetration and Victimization as Mediator Variables. *Note*. Mediation analysis was calculated with JASP [43]. Only significant loadings are presented. Loadings are unstandardized coefficients with standardized coefficients in parentheses. Gender was coded as 0 = *female* and 1 = *male*.

**Table 1 ijerph-18-10429-t001:** Descriptive Statistics and Bivariate Correlations of the Study Variables.

Variable	*M*	*SD*	01		02		03		04		05		06	07		08		09		10		11		12		13
01. global self-esteem	2.98	0.60	*0.89*																							
02. social self-esteem	3.63	0.95	**0.50**	*******	*0.70*																					
03. school performance self-esteem	3.76	0.88	**0.27**	*******	**0.26**	*******	*0.78*																			
04. body-related self-esteem	2.88	1.05	**0.37**	*******	**0.35**	*******	**0.14**	******	*0.84*																	
05. emotional self-esteem	3.34	0.98	**0.65**	*******	**0.57**	*******	**0.31**	*******	**0.55**	*******	*0.78*															
06. creative-artistic self-esteem	2.86	1.11	0.10		0.03		0.08		**0.11**	*****	**0.10**	*****	*0.83*													
07. school adjustment	4.52	1.49	**0.74**	*******	**0.74**	*******	**0.41**	*******	**0.38**	*******	**0.74**	*******	0.06	*0.95*												
08. emotional loneliness	2.78	1.05	**−0.57**	*******	**−0.53**	*******	**−0.18**	*****	**−0.28**	*******	**−0.54**	*******	−0.03	**−0.62**	*******	*0.74*										
09. social loneliness	2.50	1.18	**−0.45**	*******	**−0.60**	*******	**−0.17**	*****	**−0.31**	*******	**−0.44**	*******	−0.07	**−0.62**	*******	**0.53**	*******	*0.93*								
10. offline perpetration	1.23	0.37	**−0.17**	*****	−0.10		**−0.22**	******	−0.06		−0.05		0.08	−0.09		0.02		**0.15**	*****	*0.70*						
11. cyber perpetration	1.07	0.22	**−0.23**	******	**−0.14**	*****	**−0.29**	*******	0.06		−0.06		−0.03	**−0.16**	*****	0.10		**0.18**	*****	**0.65**	*******	*0.90*				
12. offline victimization	1.54	0.54	**−0.35**	*******	**−0.49**	*******	0.01		**−0.31**	*******	**−0.29**	*******	0.02	**−0.50**	*******	**0.38**	*******	**0.47**	*******	0.13		0.10		*0.81*		
13. cyber victimization	1.17	0.29	**−0.29**	*******	**−0.42**	*******	−0.05		**−0.17**	*******	**−0.21**	*******	−0.01	**−0.38**	*******	**0.31**	*******	**0.35**	*******	0.09		**0.18**	*****	**0.69**	*******	*0.90*

*Note.* Significant values are displayed in bold; internal consistency reliabilities (Cronbach alphas) are displayed in italics in the main diagonal. * *p* ≤ 0.05, ** *p* ≤ 0.01, *** *p* ≤ 0.001.

**Table 2 ijerph-18-10429-t002:** Results From a Factor Analysis of the Retrospective Domain-Specific Self-Esteem Items.

Retrospective Self-Esteem	Factor Loadings from Promax-Rotated Pattern Matrix					*h* ^2^
	1	2	3	4	5	
**Factor 1: emotional self-esteem (3 items)**						
06. In general, I had confidence in myself.	0.79					0.76
01. I had a good opinion of myself.	0.75					0.76
10. I often felt anxious. (R)	−0.61					0.58
**Factor 2: school performance-related self-esteem (4 items)**						
08. My teachers were satisfied with me.		0.84				0.72
13. I got poor grades because I was not working hard enough. (R)		−0.83				0.67
04. I was proud of my school performance.		0.81				0.75
11. My classes were a nuisance to me. (R) *		−0.58				0.45
**Factor 3: creative-artistic self-esteem (3 items)**						
09. I believed that I could do all kinds of artistic activities.			0.90			0.83
05. I felt I was talented in all kinds of artistic activities.			0.88			0.81
14. I felt I was better than others when performing certain artistic activities.			0.81			0.66
**Factor 4: social self-esteem (3 items)**						
07. I only felt good when I was alone. (R)				−0.88		0.76
02. In groups, I felt a sense of being alone. (R)				−0.61		0.67
15. The others doubted me. (R)				−0.52		0.65
**Factor 5: body-related self-esteem (3 items)**						
12. I thought my body was well proportioned.					0.87	0.81
16. My appearance was described as attractive.					0.84	0.74
03. I was proud of my body.					0.82	0.82
**Reliabilities (Cronbach alpha)**	0.78	0.78	0.83	0.70	0.84	
**Rotated sum of the squared loadings**	3.25	3.02	2.32	2.27	3.18	

*Note*. Listwise deletion was used for cases with missing data (*N* = 395). The extraction method was principal component factoring with an oblique (Promax with Kaiser Normalization) rotation carried out by SPSS [91]. Only factor loadings above 0.50 are shown. Reverse-scored items are denoted with an (R). Items were taken and translated from Barbot et al. [4]. * This item differs from the original version.

**Table 3 ijerph-18-10429-t003:** Prevalence and Demographic Information of the Bullying-Related Groups.

	Noninvolved Students	Pure Bullies	Pure Victims			Bully-Victims
		offline a/o cyber	only offline	only cyber	offline-cyber	offline a/o cyber
**Total sample (*N* = 439)**						
Sample size (%)	328 (74.7%)	13 (3.0%)	29 (6.6%)	16 (3.6%)	43 (9.8%)	10 (2.3%)
Number of females (%)	263 (80.2%)	3 (23.1%)	27 (93.1%)	13 (81.3%)	34 (79.1%)	7 (70.0%)
Mean age (*SD*)	23.10 (1.81)	23.62 (2.06)	22.80 (1.82)	22.81 (1.83)	22.37 (2.09)	22.80 (2.10)
**Subsample (*n* = 192)**						
Sample size (%)	128 (66.6%)	5 (2.6%)	16 (8.3%)	9 (4.7%)	31 (16.1%)	3 (1.6%)
Number of females (%)	102 (79.7%)	1 (20.0%)	14 (87.5%)	6 (66.7%)	25 (80.6%)	0 (0.0%)
Mean age (*SD*)	22.45 (2.15)	22.20 (2.77)	22.31 (2.06)	22.33 (2.12)	22.16 (2.13)	21.33 (3.51)

*Note*. A total of 20 students in the total sample (out of which 2 were also in the subsample) provided incomplete responses for at least one of the four scales (offline victimization, offline perpetration, online victimization, online perpetration), and therefore they could not be assigned to any of the bullying-related roles due to missing data.

**Table 4 ijerph-18-10429-t004:** Means and Standard Deviations in Bullying-Related Role Groups Regarding Domain-Specific Facets of Self-Esteem, Psychological School Adjustment and Loneliness.

	Noninvolved		Pure Bullies		Pure Victims						Bully-Victims		ANCOVA Results	
	Students		Offline A/O Cyber		Only Offline		Only Cyber		Offline-Cyber		Offline A/O Cyber			
**Total sample (*N* = 439)**	*M*	*SD*	*M*	*SD*	*M*	*SD*	*M*	*SD*	*M*	*SD*	*M*	*SD*	*F*(5, 385)	η_p_ ^2^
Global self-esteem	**3.09** *ce*	0.55	2.93	0.69	**2.58** *a*	0.60	2.85	0.41	**2.57** *a*	0.60	2.86	0.79	**9.07 *****	0.107
Social self-esteem	**3.85** *ce*	0.80	**4.08** *ce*	1.06	**2.89** *ab*	0.90	3.33	1.05	**2.72** *ab*	0.97	3.07	1.14	**17.39 *****	0.184
School performance self-esteem	**3.81** *b*	0.85	**2.79** *ae*	1.36	3.70	0.90	3.60	0.80	**3.92** *b*	0.72	3.39	1.04	**3.19 ****	0.040
Body-related self-esteem	**3.00** *ce*	0.97	3.14	1.38	**2.33** *ad*	0.92	**3.72** *cef*	1.11	**2.24** *ad*	0.92	**2.22** *d*	1.50	**7.72 *****	0.091
Emotional self-esteem	**3.49** *ce*	0.91	3.56	1.18	**2.61** *a*	1.03	3.42	0.97	2.70 *a*	0.93	3.00	1.04	**8.35 *****	0.098
Creative-artistic self-esteem	2.82	1.09	2.58	1.36	3.26	1.05	3.56	0.64	2.85	1.18	2.52	1.36	1.81	0.023
**Subsample (*n* = 192)**	*M*	*SD*	*M*	*SD*	*M*	*SD*	*M*	*SD*	*M*	*SD*	*M*	*SD*	*F*(5, 184)	η_p_^2^
Psychological school adjustment	**5.11** *bce*	1.20	**3.18** *a*	1.85	**3.22** *a*	1.25	4.43	1.52	**3.18** *a*	1.05	3.48	1.67	**18.86 *****	0.339
Emotional loneliness	**2.43** *ce*	0.90	3.40	1.19	**3.67** *a*	1.10	3.22	0.99	**3.56** *a*	0.90	2.78	1.26	**11.80 *****	0.242
Social loneliness	**2.05** *bce*	0.94	**3.67** *a*	1.45	**3.58** *a*	1.20	2.67	0.82	**3.48** *a*	1.07	3.00	0.88	**16.57 *****	0.309

*Note.* Significant differences are formatted in bold. The following alphanumeric indices formatted in italics indicate significant differences (at least *p* < 0.05) from the following groups: *a* noninvolved students; *b* pure bullies; *c* pure offline victims; *d* pure cyber victims; *e* pure offline-cyber victims; *f* bully-victims. A total of 20 students in the total sample (out of which 2 were also in the subsample) provided incomplete responses for at least one of the four scales (offline victimization, offline perpetration, online victimization, online perpetration), and therefore they could not be assigned to any of the bullying-related roles due to missing data. ** *p* ≤ 0.01, *** *p* ≤ 0.001.

## Data Availability

The data presented in this study are available on request from the corresponding author.

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
