# Peer review of "Perpetration and Victimization in Offline and Cyber Contexts: A Variable- and Person-Oriented Examination of Associations and Differences Regarding Domain-Specific Self-Esteem and School Adjustment"

_ijerph, 2021, doi:10.3390/ijerph181910429_

Round 1
Reviewer 1 Report
Dear authors,
I think you have done an interesting job analyzing the different domains of self-esteem in the roles of bullying and cyberbullying. However, I have some suggestions in this regard.
In the instruments, the reliability index has not been incorporated. I think it is relevant since they have adapted all the instruments to the studio sample.
In relation to Table 4, it has described the table in a very complete way in the results section, however, it would be interesting if some type of index could be added to it. This way the table would be sufficiently descriptive so as not to have to consult the results.
Regarding the discussion, I consider that the weakest section of the entire article. This section seems to repeat the results without any extra contribution. Despite the fact that the authors have done a great job in general, this section has not been sufficiently contrasted with the previous literature.
Finally, I think it would be interesting if this issue were addressed taking into account gender and age differences since self-esteem fluctuates a lot depending on these factors. I think it could be highlighted in Future Directions.
I am sure that you can improve all the aspects that I comment.
Author Response
Dear authors,
I think you have done an interesting job analyzing the different domains of self-esteem in the roles of bullying and cyberbullying. However, I have some suggestions in this regard.
Thank you for all your time and effort, we appreciate your comments.
In the instruments, the reliability index has not been incorporated. I think it is relevant since they have adapted all the instruments to the studio sample.
We agree that it is important to present information about reliabilities. Thus, we write in the section 2.3. the following “Reliabilities, means, standard deviations, and bivariate correlations are presented in Table 2”. Reliabilities are presented in Table 2 and can be found formatted in italics in the main diagonal. If it is important to you, we can again explicitly state the reliabilities for each instrument. However, since this would be redundant and require more space, we have refrained from doing so at the moment. We can of course change this if desired.
In relation to Table 4, it has described the table in a very complete way in the results section, however, it would be interesting if some type of index could be added to it. This way the table would be sufficiently descriptive so as not to have to consult the results.
We agree that our paper would benefit from this suggestion. Thus, we incorporated indices in the table to provide information regarding group differences.
Regarding the discussion, I consider that the weakest section of the entire article. This section seems to repeat the results without any extra contribution. Despite the fact that the authors have done a great job in general, this section has not been sufficiently contrasted with the previous literature.
Thank you for the positive feedback, but also for the constructive criticism. We have taken the latter as an opportunity to expand the discussion and have added several additional references to the previous literature. We think that we have been able to improve the overall paper as a result.
Finally, I think it would be interesting if this issue were addressed taking into account gender and age differences since self-esteem fluctuates a lot depending on these factors. I think it could be highlighted in Future Directions.
In our paper, we accounted for gender and age by including them as covariates in the mediational analysis as well as in the ANCOVAs. We agree, however, that a more in depth analysis of gender differences would be interesting for future research. We thus added the following sentence in the Future Direction section: “Exploring gender differences in more detail might also be of interest.”
I am sure that you can improve all the aspects that I comment.
Thank you for your helpful and insightful comments!
Christoph Burger & Lea Bachmann
Reviewer 2 Report
First of all, I would like to thank the authors for their attempts on this vulnerable topic. The paper entitled “Perpetration and victimization in offline and cyber contexts: A variable and person-oriented examination of associations and differences regarding domain-specific self-esteem and school adjustment” is genuinely fascinating. The introduction has covered all the background, problem statements, and objectives of this study. Results and discussion are well-presented and explained. I have some minor comments in the method section for the authors.
- Please add more information on data collection. When was the data collected? How long did it take for participants to fill up your questionnaire (on average)?
- It is exciting to observe that several questionnaires were used to measure variables. Some questionnaires are self-developed, such as for offline victimization and offline perpetration. How were the validity and reliability established? For the questionnaire converted into the German language, how was the reliability established?
- How was the privacy of the participants protected in this study?
- The data collection method used in this study has its limitations. People not using any online social networks are more likely to get missed, leading to a biased result that must be appropriately addressed.
Author Response
First of all, I would like to thank the authors for their attempts on this vulnerable topic. The paper entitled “Perpetration and victimization in offline and cyber contexts: A variable and person-oriented examination of associations and differences regarding domain-specific self-esteem and school adjustment” is genuinely fascinating. The introduction has covered all the background, problem statements, and objectives of this study. Results and discussion are well-presented and explained. I have some minor comments in the method section for the authors.
Thank you for spending the time to review our manuscript. We really appreciate your positive comments.
Please add more information on data collection. When was the data collected? How long did it take for participants to fill up your questionnaire (on average)?
We have now added information about when data collection took place and about the average duration of questionnaire completion.
It is exciting to observe that several questionnaires were used to measure variables. Some questionnaires are self-developed, such as for offline victimization and offline perpetration. How were the validity and reliability established? For the questionnaire converted into the German language, how was the reliability established?
The items measuring offline victimization and perpetration were formulated according to the generally accepted finding (e.g., Craig et al., 2007) that bullying usually takes different forms (e.g., physical, verbal). In formulating this scale, we also followed the response options from the established cyberbullying test (Garaigordobil, 2015).
The initial translation was done by a person with a master's degree in English language (first author, CB), and as a second step, discussed and adjusted as needed in a team consisting of the authors and three psychology students. To ensure the reliability of the instruments, Cronbach alphas were calculated (see Table 2, italicized values in the main diagonal). All reliabilities were acceptable to very good. For the self-esteem facets, factor analysis was also used to ensure construct validity (see Table 1).
How was the privacy of the participants protected in this study?
The survey was conducted using an online questionnaire with complete anonymity. Before completion, participants had to agree to an informed consent in order to participate in the study. In this they were informed that the questionnaire was completely anonymous and that all responses would be kept strictly confidential. They were also informed that the data set did not contain any information that could lead to their identification and that the data collected would be used for research purposes only. Since the data were not linked to their identity, once they had completed the online questionnaire, it was not possible to specifically delete their data record upon request, as it could no longer be linked to their identity. Participants were also informed that answering the questionnaire was entirely voluntary, that they would not be remunerated for doing so, that they were under no obligation to take part in the survey, and that they could stop answering the questionnaire at any time without giving any reason.
The data collection method used in this study has its limitations. People not using any online social networks are more likely to get missed, leading to a biased result that must be appropriately addressed.
We agree and thus we have added the following sentences in the limitation section: “Although social media penetration among Austrian youth is nearly 100% (Austrian Institute for Applied Telecommunications, 2021), the fact that the data collection took place online and was advertised on online social networks may have biased the results, as those who do not use social networks had a lower chance of being included.”
Thank you for your helpful comments and questions.
We hope to have answered all your questions to your satisfaction.
Christoph Burger & Lea Bachmann
Reviewer 3 Report
Exploring the dimensionalities of the relationship between self-esteem and bullying is a valuable addition to existing research on this subject, and considerable citation/discussion of relevant research establishes this case well. The case for eliciting post-hoc reactions to past bullying from older respondents is also well defended. Some wording is unnecessarily complex (untangle, for instance, in line 45 is clearer than disentangle). I appreciated the insights from prior research that cyberbullying is a variant of bullying (rather than its own unique phenomenon) which varies the usual power dynamics studied previously.
The sample pool is impressive with high participation, though as you note, some subsamples might not have elicited useful findings on some variables. To what degree might the homogeneous sample have affected your findings in terms of cultural norms? (You discuss education level, but not any other factors). Similarly, could eliciting participation through an online survey via online social networks have affected findings?
Author Response
Exploring the dimensionalities of the relationship between self-esteem and bullying is a valuable addition to existing research on this subject, and considerable citation/discussion of relevant research establishes this case well. The case for eliciting post-hoc reactions to past bullying from older respondents is also well defended.
Thank you for taking the time and effort to review our manuscript. We are very pleased that our research work is appreciated.
Some wording is unnecessarily complex (untangle, for instance, in line 45 is clearer than disentangle).
Thank you for the hint. We have changed disentangle to untangle. We did not notice any further language complexities when we looked through the text again. The other reviewers have not mentioned any hints in this regard either. If further changes are necessary to increase comprehension, we will be happy to make them.
I appreciated the insights from prior research that cyberbullying is a variant of bullying (rather than its own unique phenomenon) which varies the usual power dynamics studied previously.
Thank you. We agree.
The sample pool is impressive with high participation, though as you note, some subsamples might not have elicited useful findings on some variables. To what degree might the homogeneous sample have affected your findings in terms of cultural norms? (You discuss education level, but not any other factors).
We are not completely sure what the reviewer meant (we did not collect data about cultural norms in particular). However, we agree that is a valid point to make the sample homogeneity more explicit as a potential limitation. We have added the following sentences in the Limitation paragraph: “Finally, the sample consisted of German-speaking adolescents (aged 18 to 25 years) with a high level of education (about 50% had academic degrees) and, thus, results might not be generalizable to other populations. Although we did not measure socioeconomic status directly, given the high educational attainment of our participants, we cannot rule out the possibility that individuals with lower socioeconomic status, which has been shown to be a risk factor for both victimization (Jansen et al., 2012) and lower self-esteem (Haq, 2016), are underrepresented in the current sample.”
We have also incorporated the following paragraph in the discussion section: “This finding may be related to the fact that the current study involved highly educated individuals who retrospectively reported on their school years. In other words, individuals who had experienced problems in school due to victimization and dropped out of the educational system as a result are potentially underrepresented in the present sample.”
Similarly, could eliciting participation through an online survey via online social networks have affected findings?
We agree. This has also been brought up by Reviewer 2. Thus, we have added the following sentences in the limitation section: “Although social media penetration among Austrian youth is nearly 100% (Austrian Institute for Applied Telecommunications, 2021), the fact that the data collection took place online and was advertised on online social networks may have biased the results, as those who do not use social networks had a lower chance of being included.”
Thank you for your supportive comments and helpful questions.
We hope that we have addressed all of your comments to your satisfaction.
Christoph Burger & Lea Bachmann